## PERSPECTIVE

# Degeneracy in epilepsy: multiple routes to hyperexcitable brain circuits and their repair

Tristan Manfred Stöber [1,2,3], Danylo Batulin[1,4,5], Jochen Triesch[1,9],
Rishikesh Narayanan [6,9] & Peter Jedlicka [7,8,9 ✉]

Due to its complex and multifaceted nature, developing effective treatments for epilepsy is still a major challenge. To deal with this complexity we introduce the concept of degeneracy to the field of epilepsy research: the ability of disparate elements to cause an analogous function or malfunction. Here, we review examples of epilepsy-related degeneracy at multiple levels of brain organisation, ranging from the cellular to the network and systems level. Based on these insights, we outline new multiscale and population modelling approaches to disentangle the complex web of interactions underlying epilepsy and to design personalised multitarget therapies.

Degeneracy, "the ability of elements that are structurally different to perform the same function or yield the same output"[1], is a general principle, present in most complex adaptive biological systems[1,2]. Degeneracy (sometimes also called 'non-uniqueness'[3]) should be distinguished from redundancy, which results from multiple identical elements performing the same function[1,4–6], but see also ref. [7]. In contrast to redundant components, degenerate components may generate dissimilar outputs in different contexts[2,8]. Therefore, although degeneracy is sometimes called also 'partial redundancy'[9], we follow the terminology of Edelman and Gally[1] and distinguish degeneracy from redundancy.

The ubiquity of degeneracy is rooted in the advantages it entails for organisms' evolvability[9,10] and robustness[11–13]. Degeneracy allows organisms to satisfy the two seemingly contradictory goals of preserving already evolved functions and concurrently searching for and evolving new functions[10]. In this way degeneracy is linked to both robustness and innovation in evolution[1,10].

It is becoming increasingly clear that the brain also exhibits degeneracy[1,4,14–17], with similar physiological states arising from a multitude of different subcellular, cellular and synaptic mechanisms[18,19], for recent reviews see refs. [6,20,21]. However, not only similar physiological but also similar pathological brain states may arise from structurally disparate mechanisms[6,15,22,23]. Therefore, the existence of degeneracy in the brain has important implications for understanding and treating complex brain disorders such as epilepsy.

Despite considerable progress in epilepsy research, reviewed by[24–26], its complex multicausal and variable nature[27–30] has made it difficult to disentangle the underlying mechanisms. This has hindered the development of effective therapies. Apparently contradictory observations still fuel deep controversies about the origin of epilepsy. In this review, we argue that competing hypotheses can be reconciled by taking into account the concept of degeneracy. Moreover, we emphasize that degeneracy can help explain why multiple minor changes can sometimes induce

[1] Frankfurt Institute for Advanced Studies, 60438 Frankfurt am Main, Germany. [2] Institute for Neural Computation, Faculty of Computer Science, Ruhr University Bochum, 44801 Bochum, Germany. [3] Epilepsy Center Frankfurt Rhine-Main, Department of Neurology, Goethe University, 60590 Frankfurt, Germany. [4] CePTER - Center for Personalized Translational Epilepsy Research, Goethe University, 60590 Frankfurt, Germany. [5] Faculty of Computer Science and Mathematics, Goethe University, 60486 Frankfurt, Germany. [6] Cellular Neurophysiology Laboratory, Molecular Biophysics Unit, Indian Institute of Science, Bangalore 560012, India. [7] ICAR3R - Interdisciplinary Centre for 3Rs in Animal Research, Faculty of Medicine, Justus Liebig University Giessen, 35390 Giessen, Germany. [8] Institute of Clinical Neuroanatomy, Neuroscience Center, Goethe University, 60590 Frankfurt am Main, Germany. [9] These authors contributed equally: Jochen Triesch, Rishikesh Narayanan, Peter Jedlicka. ✉email: peter.jedlicka@informatik.med.uni-giessen.de

pathological phenotypes, while at other times their effects cancel, preserving healthy/physiological circuit states.

Our aim is to review examples for epilepsy-related degeneracy at three different levels of brain organization: the cellular, network and systems level. We propose that epilepsy is a group of multiscale disorders with a potential involvement of degenerate mechanisms at each level (see Fig. 1).

### Degeneracy in epilepsy at the cellular level

**Intrinsic properties and ion channel degeneracy**. It is now well established that indistinguishable single-cell and network activity can result from a variety of parameter combinations of different ion channels[18,31,32]. This phenomenon has been termed ion channel degeneracy[33,34]; for recent reviews see refs. [21,23]. Ion channel degeneracy means that "distinct channel types overlap in their biophysical properties and can thus contribute collectively to specific physiological phenotypes"[23].

There have been extensive studies of epilepsy-relevant changes in ion channels[35–37]. However, their role in the context of ion channel degeneracy and epilepsy has not yet been studied extensively. Neuronal hyperexcitability and associated epilepsy often arise as a result of combinatorial effects of multiple ion channel mutations. This has been demonstrated by an important sequencing study by Klassen et al., examining over 200 ion channel genes in epilepsy patients and healthy controls[38]. The study found a highly complex pattern of gene changes in single individuals. Even multiple nonsynonymous mutations in known monogenic risk genes for epilepsy were frequently identified in healthy individuals[38] (see Fig. 2a). This suggests that, due to ion channel degeneracy, different channel types are able to partially compensate each other's defects[23]. When a mutation affects a function of a given ion channel, other channels may rescue normal excitability[23] (see Fig. 2b). This applies both to loss-of-function as well as gain-of-function changes in ion channels. Accordingly, individuals with epilepsy typically have more than one mutation in human epilepsy genes[38].

Similarly, clinical genetic testing in patients with epilepsies that are considered to be genetic (e.g., childhood absence epilepsy or childhood epilepsy with centrotemporal spikes) often fails to detect mutations in known disease-causing genes, despite an intensive search for monogenic causes in affected families. This suggests polygenic inheritance of these classic genetic epilepsies. The difficulty in finding the contributing genes in these patients is probably related to the multicausal nature of their epilepsy.

The results of Klassen et al.[38] indicate that epilepsies are more often caused by an accumulation of mutations in multiple ion channels than by mutation in one particular ion channel[39].

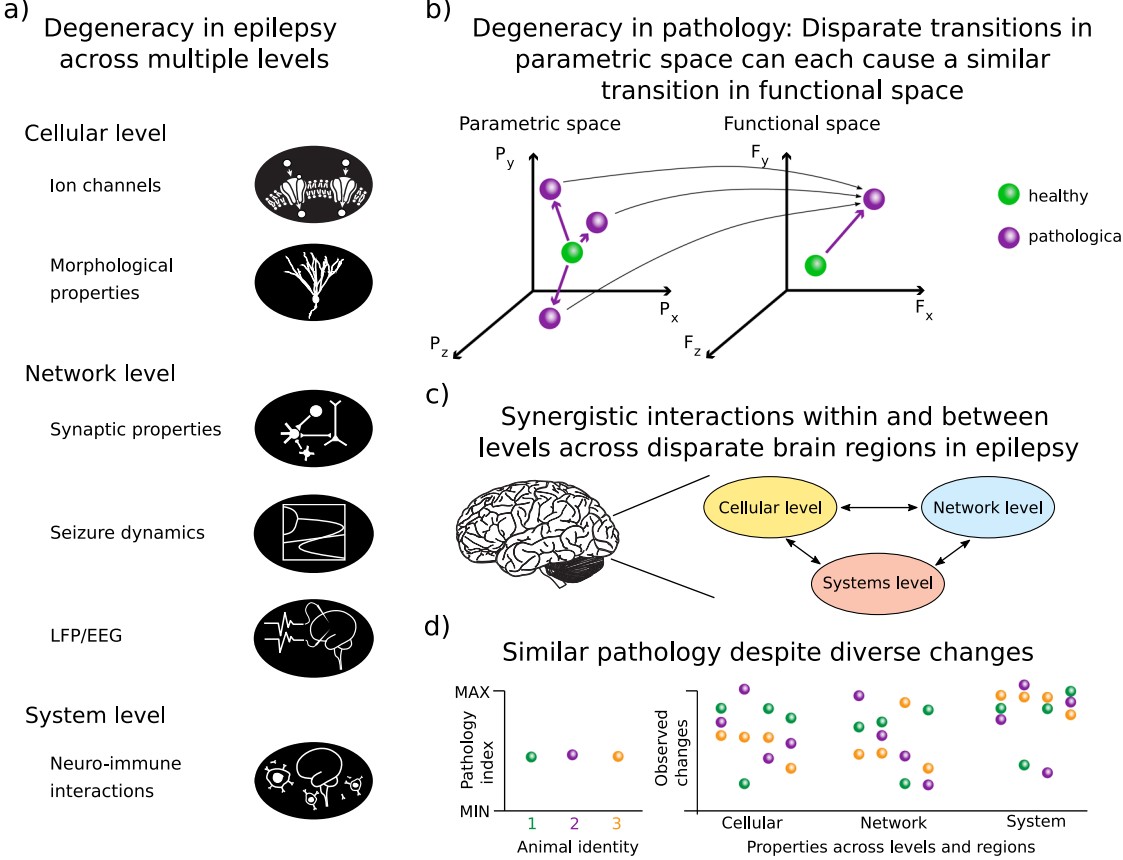

**Fig. 1 Degeneracy in the context of epilepsy spans various levels across different brain regions. a** In this article we exemplify degeneracy in epilepsy across the cellular, network and system level. Levels and sub-components are highlighted. This list reflects the organisation and scope of the article and is not intended to be exhaustive. There are likely to be other levels of organisation that express degeneracy in the context of epilepsy. **b** The concept of degeneracy, e.g. different changes leading to a similar outcome, can be visualized by a many-to-one relationship between the parametric and the functional space. Diverse changes in the parametric space can lead to a similar pathological outcome in the functional space. Green/violet spheres represent the healthy/pathological case in functional space and in the corresponding parametric space. Black thin arrows symbolise the many-to-one relationship, violet arrows the corresponding pathological transition. **c** Epilepsy is often multicausal: Pathological changes at the cellular, network and system levels can interact across multiple brain regions. **d** Thus, similar pathology indices in multiple animals can be caused by degenerate modifications of multiple properties across various levels. Some elements of **a** and **c** were adapted from[287], published under CC BY license http://creativecommons.org/licenses/by/4.0/.

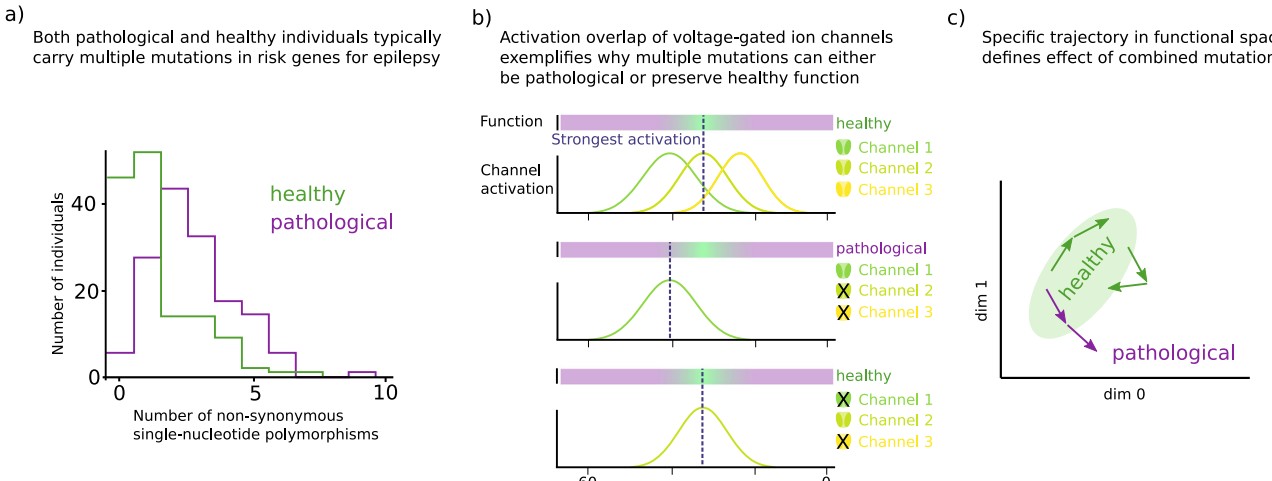

**Fig. 2 Ion channel degeneracy explains why epilepsy is typically associated with mutations in multiple risk genes. a** Genetic analysis reveals that both healthy and pathological individuals commonly carry multiple mutations in the 17 known ion channel risk genes for familial human epilepsy. Data reproduced from ref. [38]. **b** To illustrate why the combined effect of multiple mutations may sometimes, but not always, be detrimental, we create a hypothetical scenario with three voltage-gated ion channels, which are all equally effective. Let us assume, that a healthy state requires that the peak of the combined activation remains in a certain voltage band, green zone. In contrast, if the peak of the combined activation is shifted to a lower or higher voltage a pathological situation may occur, violet zone. In the upper row, all three channels are present. Due to their symmetric activation profiles, the combined activation will be strongest at the center, dashed line, and thus remains in the healthy zone. If both channel 2 and channel 3 are lost, middle row, peak activation shifts to the left and pathology ensues. In contrast, if channel 1 and channel 3 are deleted, lower row, the healthy state is maintained. Modified from ref. [21]. **c** The combined effect of multiple mutations depends on their specific trajectory in the abstract functional space: Multiple mutations can be neutral if the function simply remains in the healthy zone, upper left green trajectory, or if a detrimental mutation is compensated by a second mutation, right green trajectory. Mutations can be pathological, if they together cause the functional state to leave the healthy zone, violet trajectory.

This is supported also by computational modelling showing that combinations of small changes in the voltage activation of two or three ion channels can lead to network hyperexcitability[40]. The degeneracy of ion channels may also explain why in some individuals a lower number of mutations or hits leads to a pathological epilepsy phenotype, while other individuals with a higher number of hits appear unaffected[23]. The net effect of combined mutations will depend, first, on the initial, pre-mutation distance of a given individual (i.e. a given set of ion channel parameters) from the pathological hyperexcitable region of the possible functional space. Second, the net effect will depend on the specific trajectory in functional space determined by single or multiple mutations (Fig. 2c; see also Fig. 4). Multiple mutations in ion channels can sometimes cancel each other's effects, paradoxically preserving physiological firing behavior of affected neurons[23]. In line with this, the degeneracy between ion channel types leads to high-dimensional parameter spaces, which support robustness[21,23,41,42].

There is indirect evidence that degeneracy of dendritic ion channels is involved in epilepsy. The way in which mutations in ion channels affect the excitability of neurons is complex. This is because excitability depends not only on ion channels in the axon or the soma, but also on dendritic channels. The interaction between dendritic and somatic channels can change the way neurons respond to stimuli[37,43–48]. Mutations in some dendritic ion channels, such as A-type potassium and HCN channels, have been associated with increased excitability; however, their effects vary depending on cell type and brain region[35,49–59]. Nevertheless, experimental evidence clearly points to a significant involvement of dendritic ion channel alterations in acquired or genetic epilepsy[35,37,51,57,58] or in epilepsy-related comorbidities such as memory impairment[60].

In parallel, there is a second, epilepsy-unrelated but complementary line of direct evidence for the manifestation of ion-channel degeneracy in the emergence of characteristic somato-dendritic properties that shape normal dendritic and somatic excitability of neurons. In several mammalian principal neurons, multiple intraneuronal functional maps[61] co-exist on the same electrotonically non-compact morphology. The question is therefore whether these multiple spatial constraints on function, imposed on the same neuronal morphology, would limit the expression of degeneracy in these neurons. However, several lines of evidence, spanning different neuronal subtypes, demonstrate the emergence of similar somatodendritic functional properties despite widespread heterogeneity in the underlying ion-channel expression profiles and gating properties[20,21,48,62–67]. Thus, there are independent lines of evidence for different ion channels mediating epilepsy-associated changes to somato-dendritic excitability, and for how different combinations of ion channels might yield similar somato-dendritic functional maps. However, the convergence of these two lines of research directly investigating degeneracy in the manifestation of pathological excitability states of neuronal dendrites has been much less explored.

Nevertheless, although epilepsy research has not directly and explicitly addressed ion channel degeneracy, there is an increasing amount of indirect and implicit experimental evidence.

For example, a recent study has suggested that neural circuits express degeneracy and robustness in circuit excitability due to heterogeneous components[68]. The authors have shown that cortical neurons extracted from epileptic tissue respond more homogeneously to external stimulation than cells from healthy controls[68]. This is due to both a reduced variability in the distance between resting membrane potential and spiking threshold and a reduced dynamic range of the FI response curve. Surprisingly, neurons from epileptogenic tissue required more input to generate a low number of spikes and showed a reduced increase in firing rate for additional input. Thus, counter-intuitively the principal neurons from epileptogenic tissue were less excitable than neurons from control tissue. This suggests that hyperexcitability of individual neurons may not be necessary to induce the

hyperexcitable seizure state. Compare, for example, a right-shifted FI curve in Cunha et al.[69] with a left-shifted one in Whitebirch et al.[70]. Importantly, computer simulations of Rich et al.[68] indicated that the reduced variability in excitability could make neuronal networks more susceptible to generating synchronous epilepsy-like discharges. While it is not clear whether the reduced variability here is cause or effect of the epilepsy, these observations fit well with the concept of degeneracy. By definition, degeneracy requires heterogeneous components. Thus, reduced variability limits the ability of the system to adapt and maintain healthy function[71] and may underlie an epileptogenic transition.

Further, neuronal recordings in animal models and in the tissue of epileptic patients have generated additional data on the diversity and degeneracy of ion channel changes associated with epilepsy[36,37]. This is the case for different cell types. For example, hippocampal CA1 pyramidal cells show changes in multiple ion channel types, including upregulation of T-type calcium channels[50,72,73], downregulation of A-type potassium channels[51], and HCN channels[53]. These changes are thought to lead to increased excitability[36] in the form of enhanced firing and bursting[74]. In addition, gain-of-function mutations in the sodium channels have been identified as a cause of hyperexcitability in CA1 pyramidal neurons[75].

Similarly, changes in multiple ion channel types of neocortical pyramidal cells have been observed during epilepsy and some of them identified as a cause of pathologically increased excitability. These include epilepsy-associated changes in KCNQ2 potassium channels[76,77], BK potassium channels[78], HCN cation channels[79–81], and Nav1.6 sodium channels[82,83].

Viewed together, these studies suggest that there are multiple different routes towards hyperexcitability and its prevention or reversal in hippocampal and neocortical pyramidal neurons. Ion channel degeneracy implies that the effect of changes in one channel type will depend on the biophysical context, i.e. on the activation other intrinsic or synaptic channels in a given neuron. Indeed, for example, depending on their synaptic and intrinsic context, sodium[84], A-type potassium[33,85], SK[37,86] and HCN channels[87–90] can contribute to a suppression or an enhancement of neuronal spiking. Therefore, predicting how a specific channel alteration affects neuronal excitability will require detailed models and experimental analyses of ion channel degeneracy. We would like to encourage future studies on the role of ion channel degeneracy in epilepsy. We believe it is a promising direction for epilepsy research. In pain research, for example, recent studies have directly shown that different configurations of ion channels can lead to the hyperexcitability that underlies chronic pain[15,91,92].

**Morphological properties**. In response to epileptogenic changes in the circuitry, neurons change the structure of their dendritic trees. Altered dendritic branching has been reported in different animal models of epilepsy[93–96]. The impact of such epilepsy-related dendritic remodelling on neuronal function is poorly understood.

Computational and experimental studies have shown that morphological changes of dendrites are able to affect neuronal excitability[97–99]. Even if one keeps electrotonic properties in neuronal models unchanged, spiking behavior of neurons changes strongly depending on their size and topology[97,98,100–102]. This is the case for spiking behavior driven by somatic current injections.

In contrast, the situation may be different, if spikes are triggered by distributed synaptic inputs instead of somatic current injections. In such synaptic stimulation scenarios, spike rates, but not spike patterns, are largely independent of dendritic size and topology, provided synaptic density is preserved[103]. This has been generalized to different cell types as a universal *dendritic constancy* principle[103]. In a degeneracy-like manner, different morphological shapes help make firing rates, but not patterns, more similar.

Likewise, with active dendritic trees, diversity in neuronal morphologies has been found to be a sloppy parameter, that is, a parameter with little influence[104], in models of hippocampal CA1 pyramidal cells driven by distributed synapses[65]. CA1 pyramidal cell models remained functionally similar despite the cell-to-cell variability in dendrites because the variability was compensated by ion channel degeneracy and synaptic democracy[65]. Synaptic democracy describes a compensatory mechanism, which ensures the somatic impact remains the same irrespective of the synapse' location on the dendritic tree[105]. On the other hand, in contrast to spike rate, spiking pattern, for instance the presence of bursting in healthy[98,103] or epileptic cells[74], depends strongly on dendrite morphology and intrinsic as well as synaptic ion channels.

Overall, these studies imply that the net impact of dendritic changes in epileptic tissue depends on the interplay between morphological, synaptic and nonsynaptic ion channel changes. Indeed, this has been supported by computational modelling of epilepsy-related dendrite changes in adult-born dentate granule cells. The models have shown that isolated morphological changes, observed in epileptic animals, made neurons less excitable[106]. However, when placed in a network context with pathologically altered excitatory inputs, for example in the form of synaptic sprouting and synapse loss, altered morphologies either did not change network hyperexcitability or enhanced it[107]. Taken together, the degeneracy of morphological and biophysical parameters makes it unlikely that morphological changes of dendrites can monocausally explain epileptic hyperexcitability or its reversal.

## Degeneracy in epilepsy at the network level

**Degenerate synaptic and intrinsic properties**. Degeneracy can be found not only at the single cell level but also at the network level of synaptically connected neurons. Analogous to multiple configurations of intrinsic ion channels leading to indistinguishable electrical behavior of single neurons[108], multiple configurations of synaptic properties can support indistinguishable electrical behavior of neuronal networks[18,31,85,109,110]. In addition, circuit degeneracy is enhanced by the fact that intrinsic and synaptic channels can compensate for each other[18,31,111]. This further increases the parameter space of indistinguishable network activity. Hence both extrinsic (synaptic) as well as intrinsic properties belong to important degenerate parameters of the circuit[112]. Therefore, in principle, synaptic mechanisms can compensate for the impaired function of intrinsic ion channels and vice versa[31,111], thus increasing the robustness of the nervous system[23] by counterbalancing changes in overall synaptic drive and intrinsic excitability[113].

Degeneracy of synaptic and intrinsic properties is directly relevant in the context of epilepsy. For example, degenerate synaptic and intrinsic properties are important for the spatially and temporally sparse firing of dentate granule cells[34,42]. The sparsely active dentate gyrus is considered a protective gate for the spread of epilepsy-related hyperexcitability in the hippocampus[114–116]. The degeneracy hypothesis predicts that effective protection of the dentate gate is supported by degenerate intrinsic and synaptic mechanisms mediating default but also inducible homeostatic maintenance of firing behavior of dentate granule cells. If one set of protective mechanisms was impaired, the other set would compensate and keep the gate intact.

There exists ample experimental evidence for degenerate intrinsic and synaptic mechanisms to protect the dentate gate[117]. It is well established that even strong excitatory input from the entorhinal cortex leads only to limited activity in dentate granule cells. Their relative inertia to perforant path input is partly due to specific intrinsic biophysical properties such as hyperpolarized resting potential, spike rate adaptation and a low expression of active channels in dendrites[117,118]. Moreover, dentate granule cells have been reported to upregulate their leak (Kir, Kv1.1) and HCN channels (but see also below the discussion of the complexity of HCN channel changes) and thereby further decrease their excitability in temporal lobe epilepsy (TLE)[119–123]. This can be seen as an antiepileptic compensatory reaction[36] to proepileptic changes such as the enhancement of main excitatory perforant path synapses of dentate granule cells[124]. This suggests that a compensatory recruitment of three different ion channels can support robust maintenance of firing rate homeostasis, which would otherwise be impaired by synaptic pathology in epileptic tissue. This might partially explain the robustness of dentate granule cells against excitotoxic cell death in hippocampal sclerosis, which is associated with TLE[36]. Intriguingly, ion channels that get upregulated in dentate granule cells in epilepsy might be suitable candidates for pharmacological or genetic antiepileptic treatment[36,125].

In addition to intrinsic channels, extrinsic network mechanisms also contribute to the protection of the dentate gate. For example synaptic, phasic as well as extrasynaptic, tonic GABAergic inhibition is known to be exceptionally strong in the dentate gyrus[126]. Moreover, even synaptic inhibition itself is supported by structurally diverse mechanisms. Specifically, the existence of a stunning diversity in GABAergic interneurons of the dentate gyrus[127–131] suggests significant degeneracy in maintaining synaptic dendritic and somatic inhibition controlling the recruitment of dentate granule cells[132]. And if, despite its degeneracy and robustness, synaptic inhibition becomes impaired, tonic extrasynaptic inhibition can still provide some protection. The evidence shows namely that extrasynaptic inhibition remains preserved or even becomes enhanced in some animal models of epilepsy[133,134]. Hence, the increase in extrasynaptic inhibition in case of impaired synaptic inhibition suggests that degenerate mechanisms protect the dentate gate at the network level.

Degenerate protection mechanisms allow for multiple routes to failure. Because multiple mechanisms protect the dentate gate, there are different ways to make it fail. This is reflected in the variety of animal models of epilepsy, ranging from kindling or status-epilepticus models induced by pilocarpine, kainate or electrical stimulation[117,135] to traumatic brain injury (TBI) models[136]. Stereotypical changes in the dentate gyrus network in all these different animal models of TLE have been well characterized[137]. They include loss of mossy cells and hilar GABAergic neurons, sprouting of inhibitory synapses and changes in GABA currents and their reversal potentials, increased recurrent excitation (due to mossy fiber sprouting), enhanced adult neurogenesis and astrocytic gliosis[117,135,136,138–140].

However, despite similarities between distinct animal models of TLE, there are also differences that are still not well understood. For example, intrinsic properties of granule cells are largely unaltered in TBI models[136] in contrast to intrahippocampal kainate injection models[121]. Also changes in dendritic and somatic GABAergic synaptic inhibition are complex and difficult to interpret[139]. This is further complicated by depolarizing shifts in GABA reversal potentials caused by chloride dysregulation, reviewed in ref. [141]. We believe that future research guided by the degeneracy framework might help reconcile some controversies regarding seemingly conflicting findings of intact[142], reduced[143,144], enhanced[145] or first reduced and later enhanced[146] GABAergic inhibition in TLE. The concept of degeneracy could help elucidate how changes in inhibition in early and later stages of epileptogenesis interact with changes in excitation (including mossy fibre sprouting), cell loss and neurogenesis. The challenge is to determine, which of these changes are adaptive or maladaptive[117].

In early epileptogenesis induced by TBI (first days to weeks upon injury), an integrated picture starts emerging for synaptic and cellular changes in the dentate circuitry[136]. Interestingly, this picture seems to include degeneracy at the molecular, cellular and network levels. For example, convergent mTOR, TLR4 and VEGF receptor signaling has been found to be involved in synaptic changes, that is, enhanced excitatory AMPA currents, altered inhibition, chloride pump changes and mossy fiber sprouting, as well as in cellular changes, that is, cell loss, neurogenesis and astrogliosis (see also a recent review on neuroinflammatory cellular components of TBI-induced epileptogenesis in ref. [147]). At the network level, these synaptic and cellular changes drive jointly dentate gate disruption and contribute to epileptogenesis[136]. The details still need to be clarified but multiscale network models of the dentate gyrus emerge as a promising tool to reveal the conditions and limits under which these synaptic and cellular changes are adaptive or maladaptive[24,148–150]. Their combination with multiscale experimental approaches will help resolve controversies concerning the question to what extent do altered excitation[124] and inhibition[139], neurogenesis[151,152], mossy fiber sprouting[153–155] or mossy cell loss[156–158] contribute to dentate hyperexcitability and epileptogenesis.

The framework of degeneracy may help reconcile the 'impaired inhibition' with the 'enhanced excitation' hypotheses not only in TLE but also for example in Dravet syndrome. A Dravet-like epileptic phenotype can arise as a consequence of a Nav1.1 sodium channel deletion in GABAergic interneurons. Such an interneuron-specific Nav1.1 deletion reduces GABAergic inhibition due to impaired firing of GABAergic interneurons and is sufficient to cause Dravet-like seizures[159–162]; but see also a surprising robustness of CA1 pyramidal cells in ref. [163]. However, in line with the concept of multiple degenerate routes toward epilepsy, the Dravet-like phenotype can also result from enhanced excitability of excitatory neurons[164,165] or enhanced synaptic excitation[166] in spite of mostly intact GABAergic inhibition.

Consequently, the efficiency of personalized therapy for Dravet syndrome may depend to some extent on the underlying circuit changes, which may be cell type specific[59]. In patients with deficient sodium channels in GABAergic inhibitory neurons, a pharmacological block of sodium channels would be ineffective or even counterproductive[37,167], whereas in patients with hyperactive sodium channels in excitatory neurons it may be an effective therapy. Indeed, clinicians usually avoid giving sodium channel blockers to Dravet patients with a loss of function mutation in the SCN1A sodium channel subunit. This is also the case for those with a loss-of-function mutation in the SCN2A subunit, whereas patients with a gain-of-function mutation in the SCN2A subunit usually benefit greatly from sodium channel blockers[168,169]; see also ref. [59] for the cell type dependence of the loss/gain-of-function concepts. Similarly, in TLE patients with chloride dysregulation, leading to excitatory GABA reversal potentials, enhancement of GABAergic signaling (e.g., by benzodiazepines) might worsen rather then suppress seizures, despite pathologically diminished GABA-A conductances[170], see also ref. [171]. Thus, by drawing our attention to multiple causal links and their context, degeneracy can account for therapy failures and suggest new therapy approaches.

Degeneracy may help to explain counter-intuitive observations about inhibition and excitation not only in epileptogenesis, but also in ictogenesis. For example, a *GABAergic initiation hypothesis* has recently been proposed[172] to account for increased interneuronal activity prior to seizure onset[173–175]. This is an example of another potential route to seizures outside the typical context of excitation-inhibition imbalance. It also illustrates that although hyperexcitability of principal neurons may be a key feature of seizure-prone circuits, it is not the only pathological change leading to seizures and epilepsy. As noted above, individual excitatory neurons may not be hyperexcitable as judged from their FI curves[68] and population responses may be preceded by strong inhibition. Thus, single-neuron hyperexcitability should be distinguished from circuit hyperexcitability underlying seizures. Nevertheless, on average, excitatory and inhibitory cell activity is higher during a seizure than in the interictal state[175].

**Similar seizure dynamics despite diverse biophysical mechanisms**. There is an astounding variety of causes and features of epileptic seizures. However, surprisingly, computational analysis has revealed that seizure dynamics display certain invariant properties. Dynamical systems analysis of transitions occurring at seizure onset and offset showed that most seizures can be classified into one of 16 dynamotypes[176–178]. These dynamotypes, composed as a combination of 4 onset and 4 offset bifurcations, characterise seizure dynamics based on measurable local field potentials, with certain dynamotypes occurring more frequently and some occurring in combination[178]. The existence of multiple dynamotypes is a prime example of the degenerate nature of seizure dynamics.

Unexpectedly, similar dynamics were found in different brain regions and in three different species including human, zebrafish and mouse[176]. A combination of experiments and dynamical modelling in the study by Jirsa et al.[176] revealed that seizure dynamics share general fundamental properties[179], described in more detail below. However, this does not mean that biophysical mechanisms of seizure generation are identical across distinct brain regions and species. On the contrary, it is likely that multiple different molecular, cellular and circuit mechanisms are capable of generating the observed seizure dynamics. Therefore, the authors have proposed that "there exists a wide array of possible biophysical mechanisms for seizure genesis, while preserving central invariant properties"[176]. This proposal is in agreement with the concept of degeneracy in epilepsy.

The corresponding dynamical model, called Epileptor, captures invariant dynamical properties of seizure events using only five state variables[176], for a critical discussion see ref. [180]. A slow variable, the permittivity variable[181], determines the emergence of seizure onsets and offsets. Several different biophysical mechanisms could underlie this slow variable such as accumulation of extracellular potassium[182–184] or changes in energy metabolism[176]. Jirsa et al.[176] experimentally showed that potassium accumulation and metabolic changes (oxygen and ATP consumption) were indeed correlated with the slow onset of seizure-like events. However, other slow processes, operating on a similar time scale, e.g. short-term synaptic depression[185] or concentration changes of other ion species such as chloride[184] might also contribute to the biophysical implementation of the slow permittivity variable.

This is the case also for the remaining four faster variables. Based on electrophysiological recordings, the authors have linked the first state variable of the Epileptor to excitatory glutamatergic synaptic activity and the second state variable to inhibitory GABAergic activity[176]. Nevertheless, they showed in their own experiments that invariant dynamics of seizure-like events remain preserved even after dramatic changes of conditions such as disrupting synaptic release. This is another example showing that the Epileptor state variables, which underlie the invariant properties of simulated seizures, can be instantiated by different underlying physiological mechanisms. In agreement with this interpretation and fully in the spirit of degeneracy, the authors concluded that the above mentioned physiological correlates of the five state variables (potassium, ATP, glutamatergic and GABAergic activity) "may be only valid for (…) very specific experimental conditions" and emphasized that many other parameter configurations and trajectories could lead to the conserved system dynamics and behaviour[176], see also ref. [171]).

**Similar LFP and EEG discharges despite diversity of underlying mechanisms**. An important example for degeneracy in epilepsy at the border between macro- and micro-circuit scales is related to extracellular electrical recordings, including local field potentials (LFP), electrocorticogram (ECoG), and electro-encephalogram (EEG). These extracellular recordings constitute a crucial tool to discern physiological and pathological patterns of network activity at different spatial scales[186–194].

With specific reference to LFPs, it is clear from the analyses above that epileptic disorders are associated with changes in ion channel densities and synchrony or correlation in activity patterns. LFP and its frequency-dependent characteristics are critically reliant on single-neuron properties and on input correlations[190,194–198]. Together these imply that changes in LFP recordings are expected with epileptic disorders, which have indeed been shown across different brain regions[174,199–207]. Importantly, these observations also imply that the interpretations and analyses of extracellular recordings, their frequency-dependent properties, and spatial spread must account for pathological changes in ion channel distributions and input synchrony.

There are several lines of support for degeneracy in the generation of similar LFP and EEG patterns[20,171,194,208–210]. For instance, although EEG can be macroscopically similar across different subjects, the firing patterns and the ion-channel compositions of the distinct neurons can still be very different. In addition, LFP oscillations in the theta frequency range, which are observed in the hippocampus during exploratory behavior, have been linked both to intrinsic intra-hippocampal activation[211–216] as well as to the activation of several afferent areas including entorhinal cortex[217] and the medial septal-diagonal band of Broca[211,212,214,216]. Similar observations have been reported about the differential origins of gamma oscillations as well[216,218,219]. Thus, physiological LFP oscillations might emerge due to intra-regional but also inter-regional circuit mechanisms.

Degeneracy suggests that not only physiological patterns of activity such as theta and gamma oscillations but also pathological patterns might emerge due to multiple combinations of disparate mechanisms. For instance, hypersynchronous seizure-associated oscillations can arise from altered activity in several brain regions due to changes in disparate network components such as excitatory and inhibitory transmission or cell-specific intrinsic properties[20,171,194,208–210].

Epileptologists are becoming increasingly aware of the fact that similar macroscopic activity (as measured e.g., by EEG) can be brought about by different underlying microscopic activities. As suggested in a recent review on micro-macro relationships in seizure networks, there are possible, although not yet experimentally shown, scenarios in which similar EEG in different patients could emerge from different activities of distinct hippocampal cell

types[208]. If validated in experiments, this would be an important lesson teaching us that "ostensibly similar epilepsy expression at the macroscopic scale can originate from a variety of mechanisms at the microscopic scale"[208]. In a similar vein, the authors of a recent study on "divergent paths to seizure-like events" concluded that "ostensibly similar pathological discharges can arise from different sources"[171].

The expression of such degeneracy, and the consequent macro-micro disconnect has critical implications for how extracellular recordings are interpreted and how therapeutic targets are designed[171,208,210,220]. Failure to recognize degeneracy or the manifestation of the many-to-one mappings between neural circuit components and extracellular recordings might result in negative side effects of drugs that are ineffectual as well[208,220]. Thus, analyses and interpretation of extracellular potentials must carefully account for the manifestation of considerable heterogeneity in the parametric space of neural circuits expressing degeneracy in the generation of these potentials. Basic research at the microcircuit and cellular scales is essential to elucidate the complex one-to-many mapping between similar electrographic recordings of seizures and distinct local network mechanisms[171,208,210,220].

Together, we emphasize the need to recognize degeneracy in the emergence of disrupted extracellular signatures associated with epilepsy and the divergent paths to similar seizure events. These observations caution against extrapolation of similarities in extracellular signatures at the macroscopic scale to imply similarities of sources and mechanisms at the microscopic scale, involving disparate combinations of cellular and synaptic components[171,208–210,220].

### Degeneracy in epileptogenesis at the systems level

Similarly to the network and single-cell levels, also at the systems level, many interacting components are known to contribute to the development of epilepsy. Here we highlight the immune system, the blood brain barrier and their interactions with neural circuits.

Data from clinical and animal model studies show that similar epilepsy phenotypes can be associated with distinct underlying neuronal and inflammatory mechanisms. For example, seizures are accompanied by neuronal death in some[221] but not in other[222] animal models of epilepsy. This has led to controversies and questions whether seizures cause neuronal loss, and whether neuronal death is necessary and/or sufficient for triggering epileptogenesis[223]. Current evidence indicates that traumatic brain injury and associated neuronal death[224] are capable of triggering epilepsy development in humans[225] and model animals[226]. However, other experimental evidence suggests that epileptogenesis is possible also without apparent signs of neuronal loss when triggered by BBB breakdown[222]. Thus, neuronal loss seems sufficient but not necessary for triggering epileptogenesis. This can be understood by looking at epileptogenesis in the context of degeneracy.

One origin of degeneracy in epileptogenesis (Fig. 3) lies in the complexity of neuroimmunity[227] and its crosstalk to multiple processes in the central nervous system[228]. Recent studies emphasize the key role of the immune system in the development of epilepsy[147,228–230]. This is unsurprising given that nervous and innate immune systems have close developmental trajectories with extensive mechanistic overlap[231].

The interconnectedness of nervous and immune systems can be illustrated with functions of microglia, the major player of brain innate immunity[232,233]. This cell type is involved not only in neuroinflammation, but also in inhibiting neuronal excitability in an activity-dependent fashion[234]. Furthermore, microglia regulate the secretion of, among others, IL-1 and TNF that are known to modulate neuronal excitability and seizure threshold[228,229,235,236]. Moreover, microglia are only one among a growing list of neuroimmunity-associated players involved in epileptogenesis.

Glial activity and the associated spectrum of neuroin-flammatory reactions have been shown to be in close interplay with blood-brain barrier (BBB) permeability[237], neuronal activity[234,235], neuronal loss[223] and network reorganization [238]. Additional complexity is introduced by the fact that neuroimmune responses can be triggered not only by neurological injury, but also by downstream pathological changes which are characteristic of epilepsy and the ictal activity itself. For instance, enhanced BBB permeability was shown to be induced by epileptic seizures[239,240]. And vice versa, the spillover of brain-born substances, following BBB leakage, can cause secondary neuroinflammation[241]. In this way, seizures may develop via multiple mechanisms including BBB disruption, neuroinflammation and/or neuronal loss and network remodelling (Fig. 3).

We have recently created a phenomenological model of these major epileptogenic processes and their interactions at realistic time scales (Fig. 3)[242]. We have described the neuro-immune crosstalk in the context of neurological injury using a dynamical systems approach. In agreement with experimental data from three animal models[221,222,243–245], simulations showed that neuronal loss can be sufficient but is not necessary to drive epileptogenesis. Overall, computational modelling supports the concept that in the brain with its degenerate mechanisms, multiple different pathological mechanisms may contribute to epileptogenesis requiring different kinds of interventions for successful treatment[242]. However, there are still many unanswered questions related to cell loss. For example, the loss of certain interneuron subtypes[246] and the associated reduced inhibition and epilepsy[247] has not yet been fully understood and compared with the loss of principal neurons in the context of epileptogenesis.

### Multiscale and population modelling of degenerate circuits and epilepsy

Degeneracy is a multiscale phenomenon present at multiple levels of brain structure. Degeneracy implies that different processes on a lower level can lead to a certain phenomenon on a higher level. If phenomena on different scales interact, multiscale modelling becomes necessary. Therefore, deeper understanding of degenerate excitable systems requires multiscale computational approaches[248]. Creating models of neural circuits that can bridge two or more scales is a challenging task because of the nested hierarchy of molecular, cellular and supracellular networks with many feedforward and feedback loops between the scales. However, in well studied systems, multiscale models can provide new insights, which are relevant for epilepsy in the context of degeneracy.

A simulation study of the thalamo-cortical network in childhood absence epilepsy[249] is an intriguing example of multiscale modelling connecting the ion channel, cellular and microcircuit scales. The model showed that enhancing T-type calcium channel activation or reducing inhibitory GABA-A synaptic channel activation—in isolation as well as in combination—converted physiological network activity to seizure-like discharges. The results were in agreement with clinical observations of multiple different mutations in GABA-A receptors and T-type calcium channels in human patients and animals. The model predicted that these different mutations can lead to the same phenotype of absence epilepsy. In this way, the simulations have linked

a)

System level crosstalk of diverse epileptogenic
phenomena and spontaneous recurrent seizures

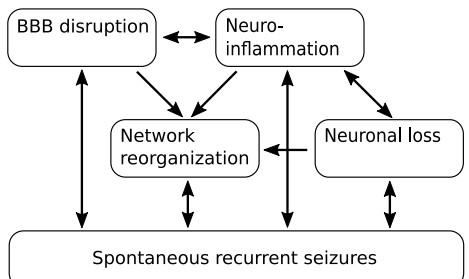

b)

Multiple distinct pathomechanisms may be sufficient but not
necessary for epileptogenesis

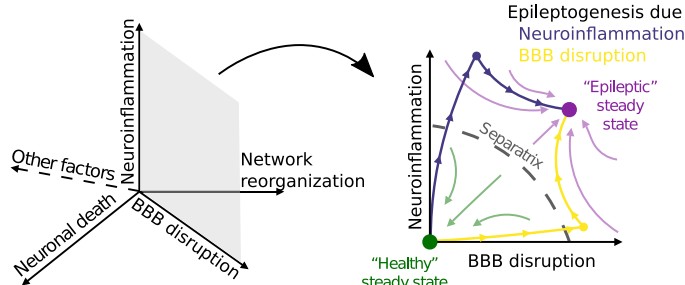

**Fig. 3 Degenerate interactions at the system level during epileptogenesis. a** Complex interactions at the system level illustrate why multiple distinct pathomechanisms may be sufficient but not necessary for epileptogenesis. **b** Experiments and simulations[243] show that different causes, such as blood-brain barrier (BBB) disruption or neuroinflammation, can lead to a similar epileptic outcome.

individual genetic variability in patients, simulated as the variability in the parameters of GABA-A and T-type calcium channels, to childhood absence epilepsy. Moreover, the model suggested plausible explanations for the failure or success of pharmacological medications targeting GABA-A and/or T-type calcium channels. Importantly, modelling predicted the necessity of multitarget therapy, simultaneously enhancing GABA-A transmission and suppressing T-type calcium channel activation, for patients with mutations in both ion channels (see also Fig. 4). Furthermore, these simulations highlight the need for personalized epilepsy therapy in patients with different genetic backgrounds. Notably, in line with the degeneracy and multicausal pathogenesis, the alterations in GABA-A and T channels do not lead to monogenic epilepsies, and therefore clinical testing does not typically reveal abnormalities in these genes in patients with childhood absence epilepsy.

Another recent multiscale simulation study, although not focusing on epilepsy, has provided novel insights on degeneracy in the context of pathological perturbations linked to hyperexcitability associated with chronic pain[109]. Using network simulations of the spinal dorsal horn, the authors have shown that under physiological conditions, similar circuit activity can emerge in different models with disparate configurations of synaptic properties. However, following identical pathological perturbations, such as a reduction of inhibition or cell type diversity, these models displayed heterogeneous circuit responses. This is in agreement with previous work showing that perturbations in degenerate, superficially similar circuits can reveal hidden variability in synaptic and intrinsic properties leading to heterogeneous responses[8,41,250,251]. Such modelling insights are highly relevant also for epilepsy. Different individuals with similar physiological circuit output may exhibit different resilience to ictogenic and epileptogenic perturbations, due to hidden variability of circuit parameters. Likewise, for the same reason, different patients with similar hyperexcitable circuit output may exhibit different susceptibility to different therapies.

In degenerate systems, instead of a single model, a large population of models with different parameter combinations is able to generate similar behavior. Consequently, in addition to multiscale modelling, degeneracy requires population modelling of neurons and neural circuits[19], also called ensemble or database modelling[20,34,48,62,65,110,252,253]. One downside of population modelling is that it is unclear which models/parameter combinations from the large theoretically possible parameter space exist in real brains. However, it is plausible to assume that evolution has removed suboptimal models from the parameter space[254].

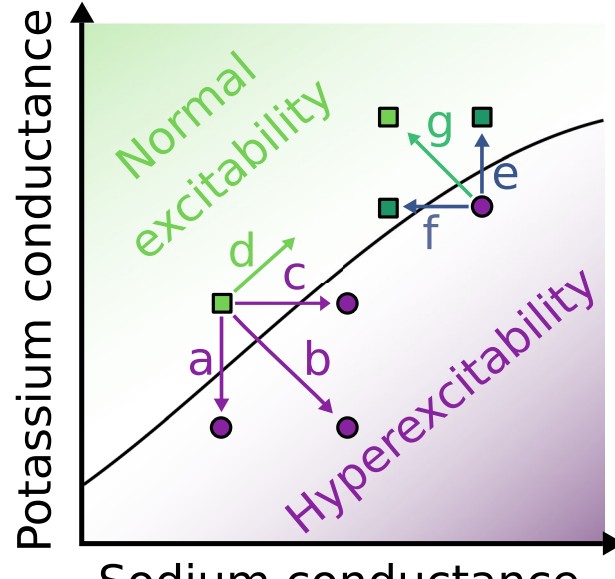

**Fig. 4 Ion channel degeneracy indicates a need for personalized single- or multitarget pharmacological therapy.** Epileptic hyperexcitability or restoration of normal excitability can be achieved by individual or combined changes in ion channels. The transition between normal excitation (green) and pathological hyperexcitability (violet) occurs when a tipping line is crossed. This transition can be induced for example by variation of sodium (horizontal axis) and potassium (vertical axis) conductance. Decrease in potassium conductance (a), increase in sodium conductance (c), or combining both (b) induces a transition from normal to pathological firing behavior of neurons. Increasing both conductances (d) maintains normal excitability. Conversely, reversal of epileptic hyperexcitability can be achieved by increasing potassium conductance (e), decreasing sodium conductance (f), or applying both changes simultaneously (g). Combined modification (g) moves the system farther away from the dangerous tipping point than isolated modification (e) or (f). For this reason, drugs might treat certain forms of epilepsy better if they modulate two or more types of ion channels simultaneously. This illustrates the potential advantage of multi-target therapy. Modified from refs. [15,261].

Indeed, evolutionary optimization principles might be useful to greatly simplify the parameter space by restricting it to the models that are Pareto optimal for the evolutionary trade-offs between multiple biologically plausible objectives[255,256]. A phenotype (cell,

circuit, organism) or its model is said to be Pareto optimal if there is no other phenotype/model that improves any of its biological objectives without worsening at least one other objective[257]. Pareto optimality for the trade-off between energy efficiency and functional effectiveness has been suggested as a guiding principle for modelling neurons and neural circuits[258,259] and for reducing their degenerate parameter space (including ion channel space) to low-dimensional manifolds[259]. Hence Pareto theory might be used to improve population modelling of healthy but also epileptic neurons and neuronal circuits.

The upside of using populations of (single scale or multiscale) models is the ability to predict novel multicausal treatment options. Currently many computer models of epilepsy treatment simulate only monocausal pharmacological effects on one type of ion channels. Using populations of conductance-based models of neurons and their circuits, implementing synaptic and intrinsic channel variability would enable predictions for a combination of multiple therapeutic targets, that is, multitarget therapy, in silico. The use of such population-based in silico models could contribute to the discovery of new antiepileptic multitarget drug cocktails[260], see Fig. 4. A recent simulation study provided a successful example for using population neuronal models as a tool for designing new multitarget drugs, which rescue pathologically increased excitability in Huntington's disease[261]. The authors called such hypothetical medicaments that optimally modulate multiple targets *holistic virtual drugs*. A similar approach could be adopted in epilepsy research for finding therapeutically efficient sets of perturbations of multiple ion channels that would switch hyperexcitable neuronal phenotypes to control phenotypes with normal excitability.

### Towards population models of human neurons and circuits in epilepsy

Recent studies of human neurons and neural circuits have begun to reveal unique properties of the human brain at the cell and circuit level. For example, human pyramidal neurons appear to exhibit enhanced dendritic compartmentalisation[262] and distinct h-channel kinetics and expression[263–265]. In addition, dendrites of human pyramidal cells show complex calcium-mediated action potentials[266], a high threshold for NMDA spikes[267] and faster backpropagation of action potentials as well as forward propagation of excitatory postsynaptic potentials[268]. The uniqueness or generality of these recently characterized properties of human neurons with respect to their animal counterparts needs to be investigated in more detail[269,270]. Nevertheless, it is plausible to assume that the cell-to-cell variability and degeneracy discovered in animal neurons[21] is also present in human neurons and their circuits.

For instance, the diversity in human h-channels (HCN channels)[264,265], see also ref. [271], might be linked to the degeneracy of electrical resonance properties[20] that are potentially important for theta oscillations in the cortex[265]. The influence of a change in HCN channel expression or kinetics on network excitability is complex due to the intricate role that HCN-mediated current ($I_h$) displays within a neuron[47,87–89,272], on the one hand depolarizing resting membrane potential[273], on the other hand decreasing input resistance and selectively diminishing dendritic excitability[57,274]. Degeneracy implies that the effects of ion channel changes may depend on the context, namely by the properties of other intrinsic and synaptic mechanisms. Therefore, compensatory or maladaptive changes in other mechanisms might explain why both loss-of-function as well as gain-of-function in $I_h$ can lead to epilepsy[90,275].

Epilepsy research has begun to benefit from the increased availability of living human brain samples. Transcriptomic,

morphological and electrophysiological data from resected human epileptic tissue started providing insights into the complex interplay between altered ion channels and morphological changes. In a recent study, such human multimodal data were obtained from living hippocampal granule cells[276]. Combined with populations of realistic cell models and network modelling, this work has provided a first step towards a better understanding of morphological and ion channel changes associated with hyperexcitability of the dentate gyrus circuitry.

### Degeneracy and its consequences for the pathophysiology and therapy of epilepsy

From a disease-etiology as well as a therapeutic standpoint, expression of degeneracy in pathological hyperexcitability states has two critical implications: (i) The source of hyperexcitability could manifest animal-to-animal and circuit-to-circuit variability, thus precluding the targeting of individual ion-channels or receptors across all animals and circuits. (ii) As hyperexcitability could emerge through disparate routes, it is equally possible that reversal of hyperexcitability could be achieved through disparate routes[15,260] (Fig. 4). The use of inhibitory receptor agonists as anticonvulsants, in many cases irrespective of the mechanistic origins of synchronous firing, constitutes an example of the latter. Thus, it is not always necessary that the reversal of neuronal hyperexcitability is achieved by reversing the mechanisms that resulted in hyperexcitability. Such reversal could be achieved through other routes without probing the mechanistic origins behind the pathological characteristic. On the other hand, under some conditions, knowledge of the precise mechanistic route towards hyperexcitability in an individual patient may be necessary for the choice of effective personalized treatment. For example, therapeutic effects of benzodiazepines and sodium channel blockers can be strongly dependent on the context of specific network pathology (see above discussion of benzodiazepines and sodium channel blockers in the section on network level degeneracy).

The recognition of degeneracy in epileptic pathology could enable simultaneous use of multiple disparate components and routes to reverse hyperexcitability. Hence there is no need for sticking to one specific drug target. In fact, to repair a failed degenerate system, it is often not enough to restore a single target mechanism (Fig. 4). One reason for this is that, due to degeneracy, many different pathologically altered mechanisms are sufficient, but usually none of them is by itself necessary for pathological malfunction[6]. In addition, a degenerate nervous system sometimes displays compensatory adaptations, which undermine therapeutic interventions focusing on a single target[15,92]. Therefore, multitarget strategies in pharmacology or in neuromodulatory stimulation might be more promising than monotarget strategies (although carrying a higher risk for side effects). In fact, many already approved antiseizure drugs affect multiple targets[277,278]. Degeneracy offers rationale also for another currently considered option of multitarget pharmacology, namely using combinations of drugs with different single targets instead of single multitarget drugs[15,278] or even using combinations of multitarget drugs. However, an intense basic and clinical research is still needed to find drug combinations with synergistic (supraadditive) therapeutic effects and low (infraadditive) toxicity[278–281]. Similarly, future clinical research is needed to provide evidence for beneficial effects of neurostimulation targeting multiple brain areas. Such evidence is currently sparse[282].

As there are many potential drug targets, the choice of specific drugs could be allowed to take advantage of circuit-specific differences in target expression profiles. For instance, if hyperexcitability is specific to a neuronal subtype in a given brain

region, this could be reversed by identifying mechanisms that are abundant in that neuronal subtype but not others. However, due to degeneracy, it is sometimes impossible to determine which neuron subtype or ion channel subtype is more crucial than others since it might depend on other, e.g., synaptic parameters[23,109].

Degeneracy of the brain creates opportunities but also challenges for precision or personalized medicine, which tries to develop therapies targeted to the specific etiology and pathophysiology of individual patients. One challenge in basic and clinical research of degeneracy is measuring multiple hyperexcitability-relevant parameters in the same individual (animal or human). Ideally, in a denegerate circuit, one would need to know most or all contributing components/mechanisms of hyperexcitability including the knowledge of most relevant hub mechanisms with highest "cruciality score" of their involvement[6]. In other words, reductionist monocausal research strategies focusing on isolated components are only partially suitable for studying complex degenerate systems[6,283]. Moreover, the conceptual framework of degeneracy with its emphasis on multitarget and multicausal thinking has consequences not only for therapy development but also for basic research and its perturbation and lesion strategies. For example, multitarget lesion experiments[6] or multi-knockout studies might be necessary to evoke dysfunction in degenerate neural or molecular networks. Measuring and perturbing multiple parameters and mechanisms simultaneously is challenging but can be facilitated by computational approaches such as multiscale and population modelling (see above).

## Epilepsy and the evolutionary costs and benefits of degeneracy

How is it possible that degeneracy makes the excitability of neural circuits robust and yet susceptible to epilepsy? We would like to argue that the evolution of the complex nervous system has led to the emergence of multiple protective mechanisms against the hyperexcitability of neural circuits but at the same time to multiple potential pathways towards the failure of this protection. In a healthy state, multiple compensatory mechanisms may act jointly with the potential to compensate for each other's failure, thereby supporting the robustness of physiological neural excitability. The compensation may be immediate ("default" or "constitutive" compensation) or it can be recruited on demand with a time delay as a form of "inducible" homeostatic plasticity ("homeostatic compensation").

Notably, degeneracy in homeostatic plasticity mechanisms has been demonstrated in visual cortex[284] and elsewhere[20,285,286,287]. The authors of visual cortex studies have explicitly stated that "multiple, partially redundant forms of homeostatic plasticity may ensure that network compensation can be achieved in response to a wide range of sensory perturbations"[284]. As we mentioned before, *partial redundancy* is a term that is sometimes used in the literature instead of degeneracy. We believe that similar research in the context of epilepsy will reveal degeneracy in homeostatic plasticity mechanisms protecting the hippocampus and other brain regions against hyperexcitability and epileptogenesis. Support for this idea comes also from computational models[288] and control theory[289], which indicate that degenerate homeostatic mechanisms provide functional benefits. For example, an effective control of neuronal firing rate, including both its mean and variance, can be achieved through the degeneracy of cooperative synaptic and intrinsic homeostatic plasticity[289].

Degeneracy in biological systems is closely linked to their complexity in terms of the number of mutually interacting individual components and mechanisms[1,290]. The greater the number of interacting mechanisms, the more complex, flexible and robust the system becomes. At the same time, the higher are also the system's energy costs linked to the extent of functional redundancy of identical components[6]. Since energy and material resources are limited, complex biological organisms display universal trade-offs between (1) functional performance, its (2) robustness and (3) flexibility and (4) energy costs[258,291]. Degeneracy facilitates the robustness and flexibility of functional behavior[10,13]. In contrast, pure redundancy increases the energy costs[6]. Through degeneracy evolution seems to have optimized biological systems for these multi-objective trade-offs[254,255,257]. A biological system tries to maximize functionality, flexibility and robustness but minimize energy expenditure[292,293]. Therefore, it is plausible that degenerate living systems evolved to become Pareto optimal for these multiple objectives. Thus, one would expect biological systems, including nervous systems[258,259,294,295], to exist close to an optimal compromise between low energy costs and high functionality (as reflected in functional effectiveness, and its robustness and flexibility). In line with these ideas, one recent study proposed that "degeneracy affords a flexibility that offsets the cost of redundancy"[296]. Another recent work indicated that heterogeneity of synaptic parameters decreases the number of synapses needed for the processing of visual inputs, resulting in a cost benefit[297]. And, intriguingly, the results of Yang et al.[295] suggested that the achievement of multiple biological goals (such as specific firing rates and energy efficiency) may only be possible with a sufficiently large diversity of interacting mechanisms (e.g., ion channels; see also[71,298]).

The conceptual link between degeneracy and evolutionary optimization opens many interesting questions. For example, counterintuitively, expressing multiple complementary protective mechanisms may paradoxically help reduce the overall cost of protection. This could be the case if activation of multiple protective mechanisms allows for optimal cost sharing among them[299]. That would open the possibility for every mechanism being expressed at the lower (and hence cheaper) end of its dynamical range[299]. In this way, the function (e.g., the homeostasis of normal excitability) would be preserved at a lower cost of its protection. Currently such optimal cost sharing is only a hypothesis that needs to be tested in experiments and simulations. On the other hand, it is also possible that the relatively high vulnerability of the nervous system with respect to hyperexcitability is a high evolutionary price that we have to pay for the enormous computational capabilities of our energy-efficient brains[292,293], which seem to operate close to criticality that maximises information-processing[300], but see also ref. [301].

Another largely unexplored area is the potential trade-off between immune defense and homeostasis of neural excitability. Immune defense of the brain is costly and under threatening conditions it might get activated at the expense of neural homeostasis[299]. Brain injuries (e.g., trauma, infection) may shift the balance and energy expenditure towards immune defense[302]. Future research may clarify whether epileptogenesis can be understood as a consequence of chronically enhanced immune defence (with its collateral damage) at the expense of suppressed excitability homeostasis.

## Summary

Developing effective treatments for epilepsy remains a challenge. The complex and multifaceted nature of this disease continues to fuel controversies about its origins. In this perspective article, we argued that conflicting hypotheses could be reconciled by considering the degeneracy of the brain, which manifests itself in multiple routes leading to similar function or dysfunction. We exemplified degeneracy at three different levels, ranging from the

cellular to the network and systems level. First, at the cellular level, we described the relevance of ion channel degeneracy for epilepsy and discussed its interplay with dendritic morphology. Second, at the network level, we provided examples of degenerate synaptic and intrinsic neuronal properties which support the robustness of neuronal networks but may also lead to diverse responses upon ictogenic and epileptogenic perturbations. Third, at the system level, we provided examples for degeneracy in the intricate interactions between the immune and nervous systems. Finally, we showed that computational approaches including multiscale and so called population (or ensemble/database) models of neurons and neural circuits might help disentangle the complex web of physiological and pathological adaptations. Such models may contribute to identifying the best personalized multitarget strategies for directing the system towards a physiological state.

**Reporting summary**. Further information on research design is available in the Nature Portfolio Reporting Summary linked to this article.

## Data availability

The graph in Fig. 2a is based on data extracted from ref. [38], Fig. 3c. See Supplementary Data 1.

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

## Acknowledgements

The work was supported by the grant LOEWE CePTER - Center for Personalized Translational Epilepsy Research (D.B., T.M.S., J.T., P.J.). D.B. is supported by the International Max Planck Research School (IMPRS) for Neural Circuits. J.T. is supported by the Johanna Quandt Foundation. R.N. is supported by a senior fellowship (IA/S/16/2/502727) from the DBT-Wellcome Trust India Alliance. P.J. is supported by the BMBF grant (No. 031L0229) and funds from the von Behring Röntgen Foundation.

## Author contributions

All authors, T.M.S., D.B., J.T., R.N., P.J., contributed to conceptualizing and writing the manuscript. R.N., D.B. and T.M.S. created the figures.

## Funding

## Competing interests

The authors declare no competing interests.

## Ethics statement

This work follows the ethical guidelines of Nature Portfolio journals.
