## [Peer Review File · Communications Biology]

Reviewers' comments:

Reviewer #1 (Remarks to the Author):

I thoroughly enjoyed reading this well-written and thoughtful review article, which synthesizes a range of studies in support of the hypothesis that degeneracy in the brain underlies the multiple pathways to seizure that typify the diverse causes of epilepsy. I thought the authors use of work outside the field of epilepsy research in which degeneracy is more well-studied to motivate future explorations was particularly insightful.

As this is a review article, my tendency as a reviewer is not to require changes that would necessitate the authors alter their overall hypotheses and writing style, as this piece should be in their voice. With that in mind, I have a few suggestions that I think are necessary to flesh out this piece (especially given some very recent studies that may have come out after the authors finished writing this manuscript), but I will present these in a way that allows the authors to integrate them into their paper as they deem appropriate. I have some other less pressing suggestions that I think would improve the piece, as well as some copy-editing issues I identified. With these addressed, I feel this already interesting manuscript will be significantly improved, and certainly worthy of publication in Communications Biology.

Necessary Additions:

1) A very recently published work in Cell Reports, entitled "Loss of neuronal heterogeneity in epileptogenic human tissue impairs network resilience to sudden changes in synchrony", compliments the perspective of this manuscript very nicely. Indeed, the concepts of heterogeneity and degeneracy echo each other, and many of the hypotheses put forth by the authors of this paper relative to degeneracy are similarly articulated relative to heterogeneity in the Cell Reports paper. Additionally, this paper includes experimental recordings from live human cortical tissue taken from patients with epilepsy similar to those suggested in the paragraph beginning on Line 171 in this piece. I fully recognize that this paper was likely published in Cell Reports after the authors finished their review, which justifies its omission; that said, I believe the manuscript will be significantly improved by incorporating a discussion of this contemporary work throughout the paper, especially given the parallels between heterogeneity and degeneracy and the experimental recordings presented in the Cell Reports paper.

2) Related to the above, one surprising finding from the recordings from epileptogenic human tissue was that these pyramidal cells were significantly LESS excitable (in terms of their FI curves) than neurons taken from non-epileptogenic tissue. This result is certainly counter-intuitive, but implies that single neuron hyperexcitability may not be strictly necessary to evoke the hyperexcitable seizure state. With this result in mind, I worry that the authors emphasis on "hyperexcitability" throughout their manuscript may be at best distracting, and at worst potentially incorrect. I feel the authors could focus on "changes associated with epilepsy" (or a similar terminology), of which hyperexcitability is one, rather than implying that hyperexcitability is the ONLY pathological change leading to seizure and epilepsy. This could be strengthened by fleshing out the discussion of alternative pathways to the seizure state, some of which were hinted at when the authors discussed the seemingly contradictory evidence regarding the

role of inhibition in seizure. A recent modeling paper published in *Frontiers in Neural Circuits* ("Inhibitory Network Bistability Explains Increased Interneuronal Activity Prior to Seizure Onset") proposed a "GABAergic initiation hypothesis" of seizure that potentially reconciles findings of increased interneuronal activity prior to seizure onset experimentally (many citations are included in that manuscript, but include Elahian et al., 2018, *Ann. Neurol.*). A discussion of this as a potential pathway to seizure outside the typical context of E-I imbalance, especially in light of the results presented in the *Cell Reports* paper, would provide a fuller picture of the multitude of potential pathways to the seizure state that would strengthen the authors argument that this is related to degeneracy.

3) As the availability of human cortical tissue increases, studies of distinctly human neurons and neural circuits are revealing unique properties of the human brain and the cell and circuit level. For instance, human neurons exhibit enhanced dendritic compartmentalization ("Enhanced Dendritic Compartmentalization in Human Cortical Neurons", *Cell*), have distinct h-channel kinetics ("Modeling Reveals Human–Rodent Differences in H-Current Kinetics Influencing Resonance in Cortical Layer 5 Neurons", *Cerebral Cortex*), and distinct expression of the h-channel ("Diversity amongst human cortical pyramidal neurons revealed via their sag currents and frequency preferences", *Nature Communications*; "h-Channels contribute to divergent intrinsic membrane properties of supragranular pyramidal neurons in human versus mouse cerebral cortex", *Neuron*). It would be interesting to see these contemporary studies cited and discussed in this review, especially given the discussion of the role of neuron morphology and the h-current in epilepsy.

4) This may be a matter of personal preference (and as such I leave the final decision making up to the writers, perhaps in light of comments from other reviewers), but I found some of the figures to be a bit "over simplified". For example, I thought Panel C in Figure 1 was perhaps superfluous, thought Figure 3 would be strengthened by including the results cited in the caption rather than "caricatures" of them, and thought Figure 4 would benefit from having the detail included in the caption for each of the numbered arrows somehow included in the figure itself. In short, I felt that the main point of some of these panels could just as easily be expressed in concise text, and thus the figures not entirely necessary. I think the manuscript would benefit from fleshing out these figures and including further details, including perhaps results from the cited papers.

Suggestions:

1) I enjoyed the authors use of the work of Steve Prescott's lab to showcase the utility of viewing a system through the lens of degeneracy. A very recent publication from their group ("Minimal requirements for a neuron to co-regulate many properties and the implications for ion channel correlations and robustness", *eLife*) extends this argument in a beautiful, mathematically motivated fashion that may be worth discussing in this manuscript.

2) I think some caveats are necessary in discussing the work of Basak and Narayanan, 2020 (Line 237). Specifically, the models presented in that work appear to be multiple models from a single morphology, rather than multiple unique models of distinct neurons. The latter would be necessary to draw

conclusions about cell-to-cell variability, and while that is partially present in the cited work it doesn't appear to be its primary focus. I think this is worth expanding on, especially given recent work focusing on the importance of the cell-to-cell variability by Eve Marder's group, as well as in the "Modeling Reveals Human–Rodent Differences in H-Current Kinetics Influencing Resonance in Cortical Layer 5 Neurons" paper mentioned above.

3) On Lines 270-271, the authors state that synaptic mechanisms "increase the robustness of the nervous system". This is intuitively true, and certainly has support from the cited work amongst others, but is it necessarily true? Could the architecture of neural circuits in some scenarios constrain, rather than expand, the potential dynamical states exhibited by the system? I believe this to be more of an open question than presented here, and think some nuance may be necessary in the portion of the paper.

4) On Line 292 the authors mention how upregulation of HCN channels can decrease the excitability of dentate granule cells. However, HCN channels are particularly funny, as there is evidence of both their over and under expression in epilepsy. This could be contextualized as additional support for the author's argument regarding degenerate pathways to the seizure state. It may be worth adding a specific discussion of this point to the manuscript if space allows, or at minimum a mention of this phenomenon in this portion of the paper so as not to accidentally imply that HCN channels only act to decrease excitability.

5) The authors do a fantastic job discussing the seminal work of Viktor Jirsa and the Epileptor, but leave out more recent work from his group ("A taxonomy of seizure dynamotypes", eLife) that speaks directly to the diversity of pathways into the seizure state. I feel this serves as further support for the argument of this manuscript and is worth discussing in some detail.

6) Related to the above: I think it's worth asking whether the "permittivity variable" in the Epileptor is perhaps too idealized, leaving room for growth in this type of modeling for direct study of what the biological analogue of this permittivity variable might be. There's also the question of the approximately periodic nature of seizures exhibited by the Epileptor, which does not approximate the reality for patients with epilepsy. Discussing questions like these, amongst others, as potential "next steps" for computational models that can yield insights into questions of degeneracy and epilepsy could be an interesting addition to this portion of the manuscript.

7) On Line 541 the authors mention the question of cell death and seizure. This question is specifically relevant in regards to the role of interneurons in epilepsy, with some studies showing death of interneurons ("Mice lacking dlx1 show subtype-specific loss of interneurons, reduced inhibition and epilepsy", Nat Neuro) and others showing other mechanisms leading to decreased inhibition ("Dendritic but not somatic gabaergic inhibition is decreased in experimental epilepsy", Nat Neuro). This could be an interesting addition to this discussion.

8) In general, I love the authors contextualization of epilepsy as a "multi-scale disorder", and wonder

whether the paper might benefit from further emphasizing this point and why it necessitates viewing the disorder through the lens of degeneracy.

9) I think the authors should add citations regarding their mention of criticality on Line 822.

Typos:

Line 71: It is becoming increasingly clear that the brain also exhibits degeneracy

Line 133: I feel the phrase "make them sick" could be interpreted as a bit crass, and should be reworked.

Line 306: in addition to intrinsic channels, extrinsic network mechanisms also

Line 321: degenerate

Line 421: Beginning this paragraph with "the authors have generated" seems a bit awkward, and I think this transition could be improved.

Line 736: delete final "the"

Reviewer #2 (Remarks to the Author):

The authors summarize and put into clear terms the previously described concept of degeneracy, the ability of diverse underlying components to generate similar outputs, as it applies to seizures and epilepsy. While many of the ideas have been stated separately (indeed, in some of the other reviews cited here), and in general are felt to be true by many in the epilepsy research community, it is a pleasant read which pulls ideas from many levels into one place and emphasizes how this concept is important to think about simultaneously at all levels.

Overall, I recommend minor revisions prior to acceptance, one of which would be to simplify the sentence structure and language throughout to increase accessibility and clarity. A few sections could benefit from additional expansion of specific examples (as was done beautifully in the section on the dentate gyrus), and I have a few specific comments below. Otherwise, a nicely organized paper with many ideas that will resound with researchers and clinicians alike on this topic.

Major:

In section 4, it would be worth pointing out that several known 'genetic' epilepsies (childhood epilepsy with centrotemporal spikes, absence, and other generalized epilepsies) often are not found to have mutations in known disease causing genes during clinical genetic testing, despite significant effort looking for monogenic causes in affected families, suggesting that polygenic inheritance for these classically genetic epilepsies is important. You imply this is the case, but the difficulty in finding causative genes in these cases is known well clinically and underscores your point in this section. On p27, likewise, it should be also noted that while these genes have been noted to contribute to susceptibility, they are not monogenic and most often during clinical testing, people with childhood absence do not have clear abnormalities in these genes which makes direct clinical application of specific dysfunction in absence at this time difficult.

The discussion of the somatodendritic and spatial differences in ion channels is cursory, consisting mainly of citations where information can be found. Can more information be included to specify what changes have, for example, been seen? A bit of interpretation of the cited literature would be helpful here.

In section 5, the molecular changes in various signaling pathways come up, and given this, it would seem that molecular (non-ion channel) changes should likely be included in part 4.1 and also under the cellular section in figure 1.

p18.388-392 it should be pointed out here in support of this point that sodium channel agents are typically avoided clinically in most Dravet patients with loss of function in *SNC1A* while in most *SCN2A* patients with gain of function, high dose sodium channel agents are found to be particularly effective.

p29.658 please define Pareto before starting to use it here. Depending on your audience, you may lose people without some explanation.

p30.677-678 please expand upon what this reference did. What is meant by 'holistic virtual drugs'?

Minor:

In Figure 1, please ensure that it is noted that this is not an exhaustive list. Likely many additional parameters (e.g. additional signaling pathways) could be added. I think 1D is the most helpful in expressing the concept, while 1B could be clarified more and 1C seems a bit unnecessary (though a nice diagram).

p34.765-772: this section seems out of place, and could be moved above to the network section

minor grammatical errors: e.g. p3.34 fuels; p3.39 levels, p3.41 examples of or evidence for... that support...also lead, p3.44 examples of or evidence for, p3.45 immune and nervous systems
p8.149-p9.158-- awkward paragraph, consider rewording

p12.236-237 please define synaptic democracy

p13.263-264 awkward sentence consider rewording

p16.344 reconcile

p17.348 remove 'the' prior to TLE

p19.402 There are, p19.409 these kinds of dynamics were or this kind of dynamic was

p20.447 lead to conserved (remove 'the')

p22.499 what is meant by 'mesoscopic'?

p25.586 regulate, p25.572 and the associated

p26.576 ...complexity is introduced by the fact...not only by neurological injury, but also from downstream

p28.632 and p29.653 too many commas

p29.655 evolution has removed

p29.667 using multiple models, p29.670 channel. Using multiple conductance-based... p29.671 remove comma

p33.748 for therapy... for basic research (remove 'the'), p33.751 evoke dysfunction in degenerate...

p34.783-787 remove "system's" or reword the sentence, awkward

p34.787 performing one function, A (redundancy), then... multiple structurally different... This sentence is still very unclear to me, consider removing this and the following sentence altogether-- the point has been made already.

I would also recommend removing many parenthetical extra words and just leaving them in the sentence, as this is a distracting way of writing for many of the cases in this paper.

1841 **Revision**

1842 We thank the reviewers for the constructive feedback. In our opinion, incorpo-
1843 rating the suggested changes greatly improved the manuscript.

1844 **Reviewer 1**

1845 I thoroughly enjoyed reading this well-written and thoughtful review article,
1846 which synthesizes a range of studies in support of the hypothesis that degen-
1847 eracy in the brain underlies the multiple pathways to seizure that typify the
1848 diverse causes of epilepsy. I thought the authors use of work outside the field of
1849 epilepsy research in which degeneracy is more well-studied to motivate future
1850 explorations was particularly insightful.

1851 As this is a review article, my tendency as a reviewer is not to require
1852 changes that would necessitate the authors alter their overall hypotheses and
1853 writing style, as this piece should be in their voice. With that in mind, I have
1854 a few suggestions that I think are necessary to flesh out this piece (especially
1855 given some very recent studies that may have come out after the authors finished
1856 writing this manuscript), but I will present these in a way that allows the authors
1857 to integrate them into their paper as they deem appropriate. I have some
1858 other less pressing suggestions that I think would improve the piece, as well as
1859 some copy-editing issues I identified. With these addressed, I feel this already
1860 interesting manuscript will be significantly improved, and certainly worthy of
1861 publication in Communications Biology.

1862 Necessary Additions:

- 1863 1. A very recently published work in Cell Reports, entitled "Loss of neu-
1864 ronal heterogeneity in epileptogenic human tissue impairs network re-
1865 siliance to sudden changes in synchrony", compliments the perspective
1866 of this manuscript very nicely. Indeed, the concepts of heterogeneity and

1867 degeneracy echo each other, and many of the hypotheses put forth by
1868 the authors of this paper relative to degeneracy are similarly articulated
1869 relative to heterogeneity in the Cell Reports paper. Additionally, this pa-
1870 per includes experimental recordings from live human cortical tissue taken
1871 from patients with epilepsy similar to those suggested in the paragraph
1872 beginning on Line 171 in this piece. I fully recognize that this paper was
1873 likely published in Cell Reports after the authors finished their review,
1874 which justifies its omission; that said, I believe the manuscript will be
1875 significantly improved by incorporating a discussion of this contemporary
1876 work throughout the paper, especially given the parallels between hetero-
1877 geneity and degeneracy and the experimental recordings presented in the
1878 Cell Reports paper.

1879 Thank you very much for pointing us to the interesting work by Rich et
1880 al. 2022. We agree that degeneracy and heterogeneity are closely related.
1881 Indeed, by definition, only a system with heterogeneous components can
1882 express degeneracy. Thus, we have added a short description of their
1883 findings in a separate paragraph spanning lines 190 to 211.

1884 2. Related to the above, one surprising finding from the recordings from
1885 epileptogenic human tissue was that these pyramidal cells were signifi-
1886 cantly LESS excitable (in terms of their FI curves) than neurons taken
1887 from non-epileptogenic tissue. This result is certainly counter-intuitive,
1888 but implies that single neuron hyperexcitability may not be strictly neces-
1889 sary to evoke the hyperexcitable seizure state. With this result in mind, I
1890 worry that the authors emphasis on "hyperexcitability" throughout their
1891 manuscript may be at best distracting, and at worst potentially incorrect.
1892 I feel the authors could focus on "changes associated with epilepsy" (or
1893 a similar terminology), of which hyperexcitability is one, rather than im-

1894 plying that hyperexcitability is the ONLY pathological change leading to
1895 seizure and epilepsy. This could be strengthened by fleshing out the dis-
1896 cussion of alternative pathways to the seizure state, some of which were
1897 hinted at when the authors discussed the seemingly contradictory evidence
1898 regarding the role of inhibition in seizure. A recent modeling paper pub-
1899 lished in *Frontiers in Neural Circuits* ("Inhibitory Network Bistability Ex-
1900 plains Increased Interneuronal Activity Prior to Seizure Onset") proposed
1901 a "GABAergic initiation hypothesis" of seizure that potentially reconciles
1902 findings of increased interneuronal activity prior to seizure onset exper-
1903 imentally (many citations are included in that manuscript, but include
1904 Elahian et al., 2018, *Ann. Neurol.*). A discussion of this as a potential
1905 pathway to seizure outside the typical context of E-I imbalance, especially
1906 in light of the results presented in the *Cell Reports* paper, would pro-
1907 vide a fuller picture of the multitude of potential pathways to the seizure
1908 state that would strengthen the authors argument that this is related to
1909 degeneracy.

1910 Thank you. We have included this point ("single neuron hyperexcitability
1911 may not be strictly necessary to evoke the hyperexcitable seizure state") in
1912 the revised manuscript - see also l. 190 to 211. We now distinguish between
1913 single-neuron and circuit hyperexcitability (the latter one is present in
1914 our title). We have now cited the Elahian paper and the "GABAergic
1915 initiation hypothesis" paper. We have added a short paragraph, in which
1916 we argue that although *hyperexcitability* may be a key feature of seizure-
1917 prone circuits, hyperexcitability is not the only pathological change leading
1918 to seizure and epilepsy. We fully acknowledge that individual neurons may
1919 not be hyperexcitable as judged from their FI curves (Rich et al., 2022) and
1920 that populations responses may be preceded by strong inhibition. We also

1921 mention, that, however, on average, activity of excitatory and inhibitory
1922 cells during a seizure is higher compared to the interictal state. This is
1923 for example vividly illustrated in Figures 3-5 of Elahian et al. (2018). See
1924 lines 452 to 466.

1925 3. As the availability of human cortical tissue increases, studies of distinctly
1926 human neurons and neural circuits are revealing unique properties of the
1927 human brain and the cell and circuit level. For instance, human neurons
1928 exhibit enhanced dendritic compartmentalization ("Enhanced Dendritic
1929 Compartmentalization in Human Cortical Neurons", Cell), have distinct
1930 h-channel kinetics ("Modeling Reveals Human-Rodent Differences in H-
1931 Current Kinetics Influencing Resonance in Cortical Layer 5 Neurons",
1932 Cerebral Cortex), and distinct expression of the h-channel ("Diversity
1933 amongst human cortical pyramidal neurons revealed via their sag cur-
1934 rents and frequency preferences", Nature Communications; "h-Channels
1935 contribute to divergent intrinsic membrane properties of supragranular
1936 pyramidal neurons in human versus mouse cerebral cortex", Neuron). It
1937 would be interesting to see these contemporary studies cited and discussed
1938 in this review, especially given the discussion of the role of neuron mor-
1939 phology and the h-current in epilepsy.

1940 In line with the suggestion of the reviewer, we have added a paragraph
1941 citing and briefly discussing the contemporary studies of within-species as
1942 well as inter-species diversity with respect to human neuronal properties
1943 including h-channels. See line number: 767.

1944 4. This may be a matter of personal preference (and as such I leave the final
1945 decision making up to the writers, perhaps in light of comments from other
1946 reviewers), but I found some of the figures to be a bit "over simplified".
1947 For example, I thought Panel C in Figure 1 was perhaps superfluous,

1948 thought Figure 3 would be strengthened by including the results cited
1949 in the caption rather than "caricatures" of them, and thought Figure 4
1950 would benefit from having the detail included in the caption for each of
1951 the numbered arrows somehow included in the figure itself. In short, I
1952 felt that the main point of some of these panels could just as easily be
1953 expressed in concise text, and thus the figures not entirely necessary. I
1954 think the manuscript would benefit from fleshing out these figures and
1955 including further details, including perhaps results from the cited papers.

1956 Thank you very much for your suggestions.

1957 We understand that Figure 1C highlights a somewhat trivial point. Patho-
1958 logic processes in the brain interact across multiple levels. However, since
1959 this point is important for Fig. 1D, we would prefer to keep it.

1960 Regarding Figure 3, we thoroughly considered to include original results
1961 from the Batulin et al. (2022) article. However, we decided against it.
1962 Adding any original figure, such 7B, would require to explain the model
1963 in more detail and to provide information regarding the nature of fixed
1964 points, which we consider beyond the scope of this work.

1965 Suggestions:

- 1966 1. I enjoyed the authors use of the work of Steve Prescott's lab to showcase
1967 the utility of viewing a system through the lens of degeneracy. A very
1968 recent publication from their group ("Minimal requirements for a neu-
1969 ron to co-regulate many properties and the implications for ion channel
1970 correlations and robustness", eLife) extends this argument in a beautiful,
1971 mathematically motivated fashion that may be worth discussing in this
1972 manuscript.

1973 We thank the reviewer for pointing us to this elegant work by the Prescott
1974 laboratory. We have now included this study in the section where we refer

1975 to Pareto optimality where co-regulation of multiple properties is the core
1976 argument. Specifically, we have included this reference in the section where
1977 we address the question of achieving multiple objectives including energy
1978 efficiency and functional robustness, a point that is particularly emphasized
1979 by Yang et al., eLife, 2022 as well (Line numbers: 927, 932).

1980 2. I think some caveats are necessary in discussing the work of Basak and
1981 Narayanan, 2020 (Line 237). Specifically, the models presented in that
1982 work appear to be multiple models from a single morphology, rather than
1983 multiple unique models of distinct neurons. The latter would be neces-
1984 sary to draw conclusions about cell-to-cell variability, and while that is
1985 partially present in the cited work it doesn't appear to be its primary
1986 focus. I think this is worth expanding on, especially given recent work
1987 focusing on the importance of the cell-to-cell variability by Eve Marder's
1988 group, as well as in the "Modeling Reveals Human-Rodent Differences in
1989 H-Current Kinetics Influencing Resonance in Cortical Layer 5 Neurons"
1990 paper mentioned above.

1991 Please note that work of Basak and Narayanan, 2020 involves multiple
1992 morphologies. However, to consider this point of the reviewer, we have
1993 now included the suggested paper (Rich et al., 2021) and mention the link
1994 to cell-to-cell variability. See line numbers: 767, 782.

1995 3. On Lines 270-271, the authors state that synaptic mechanisms "increase
1996 the robustness of the nervous system". This is intuitively true, and cer-
1997 tainly has support from the cited work amongst others, but is it necessarily
1998 true? Could the architecture of neural circuits in some scenarios constrain,
1999 rather than expand, the potential dynamical states exhibited by the sys-
2000 tem? I believe this to be more of an open question than presented here,
2001 and think some nuance may be necessary in the portion of the paper.

2002 Please note that our argument here was on how synaptic and intrinsic
2003 mechanisms could potentially compensate for each others impairments in
2004 achieving robust functions. We do not intend to convey that the architec-
2005 ture of neural circuits expand the potential dynamical states exhibited by
2006 the system.

2007 One way to expand on this would be looking at the traditional idea of E-I
2008 balance instead from the perspective of balance between E-I and intrinsic
2009 excitability. If subthreshold voltage responses are the homeostatic control
2010 variable, then the output voltage would be a function of the net excita-
2011 tory inputs, the net inhibitory inputs, and the subthreshold gain (input
2012 resistance, input impedance, temporal summation, etc.) of the system.
2013 If firing rate of the neuron is the homeostatic control variable, then the
2014 net firing rate gain of the system plays a critical role rather than just the
2015 excitatory and inhibitory drive. As subthreshold or firing rate gain are
2016 referred to as intrinsic excitability, the overall output of a neuron could
2017 be considered as a balance between E-I-IE, where IE stands for intrin-
2018 sic excitability (expanded in Seenivasan and Narayanan, Curr. Opinion
2019 Neurobiology, 2022). Our argument here is that impairments to synaptic
2020 properties could be compensated by counterbalancing changes in intrinsic
2021 excitability and vice versa.

2022 To emphasize these points, we have now added the point about balance
2023 between excitatory and inhibitory synaptic inputs as well as intrinsic ex-
2024 citability in the relevant location (Line number: 309).

2025 4. On Line 292 the authors mention how upregulation of HCN channels can
2026 decrease the excitability of dentate granule cells. However, HCN channels
2027 are particularly funny, as there is evidence of both their over and under
2028 expression in epilepsy. This could be contextualized as additional support

2029 for the author's argument regarding degenerate pathways to the seizure
2030 state. It may be worth adding a specific discussion of this point to the
2031 manuscript if space allows, or at minimum a mention of this phenomenon
2032 in this portion of the paper so as not to accidentally imply that HCN
2033 channels only act to decrease excitability.

2034 Thank you for the suggestion. We have now added a brief discussion of
2035 HCN channels in the new paragraph, in which we also discuss human
2036 neuronal data - and we refer now to this paragraph at the text location
2037 where we mention the upregulation of HCN channels (Line number: 782).

2038 5. The authors do a fantastic job discussing the seminal work of Viktor Jirsa
2039 and the Epileptor, but leave out more recent work from his group ("A tax-
2040 onomy of seizure dynamotypes", eLife) that speaks directly to the diversity
2041 of pathways into the seizure state. I feel this serves as further support for
2042 the argument of this manuscript and is worth discussing in some detail.
2043 Thank you for pointing us to this new work. Following your suggestion, we
2044 included a short description of this work and the corresponding reference.
2045 See l. 471.

2046 6. Related to the above: I think it's worth asking whether the "permittivity
2047 variable" in the Epileptor is perhaps too idealized, leaving room for growth
2048 in this type of modeling for direct study of what the biological analogue
2049 of this permittivity variable might be. There's also the question of the ap-
2050 proximately periodic nature of seizures exhibited by the Epileptor, which
2051 does not approximate the reality for patients with epilepsy. Discussing
2052 questions like these, amongst others, as potential "next steps" for compu-
2053 tational models that can yield insights into questions of degeneracy and
2054 epilepsy could be an interesting addition to this portion of the manuscript.
2055 Thanks, we have tried to insert these points by adding that "Direct exper-

2056 imental studies combined with adjusted, more detailed Epileptor models
2057 might help clarify the specific biological implementations of the permit-
2058 tivity variable under different pathological conditions.” and ”the epileptor
2059 model could be improved to account for the stochastic nature of seizures
2060 in patients with epilepsy.” However, the flow of the text got interrupted so
2061 we decided not to include it after all. We believe that these two additions,
2062 although being valid and interesting, are not crucial for the ”degeneracy”
2063 message of the Epileptor section. However, we pointed the reader to crit-
2064 ical discussions of the Epileptor on l. 491.

2065 7. On Line 541 the authors mention the question of cell death and seizure.
2066 This question is specifically relevant in regards to the role of interneu-
2067 rons in epilepsy, with some studies showing death of interneurons (”Mice
2068 lacking dlx1 show subtype-specific loss of interneurons, reduced inhibition
2069 and epilepsy”, Nat Neuro) and others showing other mechanisms leading
2070 to decreased inhibition (”Dendritic but not somatic gabaergic inhibition is
2071 decreased in experimental epilepsy”, Nat Neuro). This could be an inter-
2072 esting addition to this discussion. To follow your suggestion, we included
2073 the issue related to the death of interneurons and included the mentioned
2074 references. See l. 666.

2075 8. In general, I love the authors contextualization of epilepsy as a ”multi-scale
2076 disorder”, and wonder whether the paper might benefit from further em-
2077 phasizing this point and why it necessitates viewing the disorder through
2078 the lens of degeneracy.

2079 Thank you very much. We are glad you enjoyed this aspect. We hesi-
2080 tate to expand it by adding more sentences - mainly due to the already
2081 considerable length of the review.

2082 9. I think the authors should add citations regarding their mention of criti-
2083 cality on Line 822.

2084 Thanks, we have added two citations and toned down the sentence. See
2085 line number 954.

2086 Typos:

2087 1. Line 71: It is becoming increasing clear that the brain also exhibits degen-
2088 eracy Thank you for noting typos here! We have corrected the position of
2089 *also* in the manuscript. See line 70.

2090 2. Line 133: I feel the phrase "make them sick" could be interpreted as a bit
2091 crass, and should be reworked. Thank you, replaced by *other individuals*
2092 *with a higher number of hits appear unaffected*. See line 137.

2093 3. Line 306: in addition to intrinsic channels, extrinsic network mechanisms
2094 also Corrected. See line 350.

2095 4. Line 321: degenerate Corrected. See line 365.

2096 5. Line 421: Beginning this paragraph with "the authors have generated"
2097 seems a bit awkward, and I think this transition could be improved. We
2098 have simplified the phrase to *The corresponding dynamical model, called*
2099 *Epileptor, captures* We hope this makes it less awkward. See line 491.

2100 6. Line 736: delete final "the" Thank you. We realized there were a number
2101 of superfluous *the* in the subsection. We removed them.

2102 **Reviewer 2**

2103 The authors summarize and put into clear terms the previously described con-
2104 cept of degeneracy, the ability of diverse underlying components to generate
2105 similar outputs, as it applies to seizures and epilepsy. While many of the ideas

2106 have been stated separately (indeed, in some of the other reviews cited here),
2107 and in general are felt to be true by many in the epilepsy research community, it
2108 is a pleasant read which pulls ideas from many levels into one place and empha-
2109 sizes how this concept is important to think about simultaneously at all levels.
2110 Overall, I recommend minor revisions prior to acceptance, one of which would
2111 be to simplify the sentence structure and language throughout to increase ac-
2112 cessibility and clarity. A few sections could benefit from additional expansion of
2113 specific examples (as was done beautifully in the section on the dentate gyrus),
2114 and I have a few specific comments below. Otherwise, a nicely organized paper
2115 with many ideas that will resound with researchers and clinicians alike on this
2116 topic.

2117 Major:

2118 1. In section 4, it would be worth pointing out that several known 'genetic'
2119 epilepsies (childhood epilepsy with centrotemporal spikes, absence, and
2120 other generalized epilepsies) often are not found to have mutations in
2121 known disease causing genes during clinical genetic testing, despite signif-
2122 icant effort looking for monogenic causes in affected families, suggesting
2123 that polygenic inheritance for these classically genetic epilepsies is impor-
2124 tant. You imply this is the case, but the difficulty in finding causative
2125 genes in these cases is known well clinically and underscores your point
2126 in this section. On p27, likewise, it should be also be noted that while
2127 these genes have been noted to contribute to susceptibility, they are not
2128 monogenic and most often during clinical testing, people with childhood
2129 absence do not have clear abnormalities in these genes which makes direct
2130 clinical application of specific dysfunction in absence at this time difficult.

2131 Thanks. We have followed these suggestions. See l. 123 and 702.

2132 2. The discussion of the somatodendritic and spatial differences in ion chan-

2133 nels is cursory, consisting mainly of citations where information can be
2134 found. Can more information be included to specify what changes have,
2135 for example, been seen? A bit of interpretation of the cited literature
2136 would be helpful here.

2137 We thank the reviewer for their suggestion. We have now added a short
2138 paragraph providing context to the cited literature at the specific location
2139 where we discuss somatodendritic gradients and ion-channel degeneracy.
2140 Line number: 167–184.

2141 3. In section 5, the molecular changes in various signaling pathways come up,
2142 and given this, it would seem that molecular (non-ion channel) changes
2143 should likely be included in part 4.1 and also under the cellular section in
2144 figure 1.

2145 While we thank the reviewer for the suggestion and agree that in many
2146 of the described process molecular changes are involved, we are somewhat
2147 hesitant to extend the manuscript to describe such changes in more detail.
2148 The manuscript is already quite long. To accord with the structure of the
2149 manuscript, describing them properly would also necessitate to create a
2150 new subsection. Figure 1A reflects this structure and since its meant to
2151 be a non-exhaustive list, see comment below, we only include subsections.
2152 Nevertheless, if the reviewer insists we can include a new subsection with
2153 molecular (non-ion channel) changes.

2154 4. p18.388-392 it should be pointed out here in support of this point that
2155 sodium channel agents are typically avoided clinically in most Dravet pa-
2156 tients with loss of function in SNC1A while in most SCN2A patients with
2157 gain of function, high dose sodium channel agents are found to be partic-
2158 ularly effective. Thanks! Done. See l. 438

2159 5. p29.658 please define Pareto before starting to use it here. Depending on
2160 your audience, you may lose people without some explanation. Thanks!
2161 Done. See l. 738

2162 6. p30.677-678 please expand upon what this reference did. What is meant
2163 by 'holistic virtual drugs'? We have added a brief explanation. See l. 759

2164 Minor:

2165 1. In Figure 1, please ensure that it is noted that this is not an exhaustive list.
2166 Likely many additional parameters (e.g. additional signaling pathways)
2167 could be added. I think 1D is the most helpful in expressing the concept,
2168 while 1B could be clarified more and 1C seems a bit unnecessary (though
2169 a nice diagram).

2170 Thank you. To clarify that the list in 1A is non-exhaustive we added two
2171 sentences *This list reflects the organization and scope of the article and is*
2172 *not meant to be exhaustive. Additional levels of organization expressing*
2173 *degeneracy in the context of epilepsy likely exist.*

2174 We simplified 1B to facilitate understanding.

2175 With a simplified 1B, we prefer to keep 1C to make sure that the reader
2176 can follow why we talk about different levels (cellular, network, system)
2177 in 1D. Caption of 1C now reads *Epilepsy is often multicausal: Pathologi-*
2178 *cal changes at the cellular, network and system level may interact across*
2179 *multiple brain regions..*

2180 2. p34.765-772: this section seems out of place, and could be moved above
2181 to the network section We would prefer to keep this section at the end -
2182 to put the concept of degeneracy into an evolutionary context. To make
2183 this clearer, we have changed the title of the section to "Epilepsy and the

2184 evolutionary costs and benefits of degeneracy”. We have tried to make
2185 the transition from the previous section smoother. See line 882.

2186 3. minor grammatical errors: e.g. p3.34 fuels; p3.39 levels, p3.41 examples
2187 of or evidence for... that support...also lead, p3.44 examples of or evi-
2188 dence for, p3.45 immune and nervous systems p8.149-p9.158– awkward
2189 paragraph, consider rewording Thank you.

2190 • **p3.34 fuels p3.39 levels** Unclear where the error is.

2191 • **p3.39 levels** We think the singular form is correct, because it is just
2192 a short version of *ranging from the cellular level to the network level*
2193 *and systems level*

2194 • **p3.41 examples of or evidence for... that support...also lead**
2195 Corrected to: *Second, at the network level, we provide examples of*
2196 *degenerate synaptic and intrinsic neuronal properties which support*
2197 *the robustness of neuronal networks but may also lead to diverse re-*
2198 *sponses upon ictogenic and epileptogenic perturbations.* See line 40.

2199 • **p3.45 immune and nervous systems** Thanks, we have corrected
2200 it.

2201 • **p8.149-p9.158– awkward paragraph, consider rewording** We
2202 reformatted the respective paragraph. See l. 149 to 166.

2203 4. p12.236-237 please define synaptic democracy Thanks, a definition has
2204 been added. See line 277.

2205 5. p13.263-264 awkward sentence consider rewording Thank you, we agree.
2206 Sentence has been reworded and we hope its better now. See line 304.

2207 6. p16.344 reconcile Thanks. Corrected, see line 388.

2208 7. p17.348 remove 'the' prior to TLE Thanks. Corrected.

2209 8. p19.402 There are, p19.409 these kinds of dynamics were or this kind of
2210 dynamic was

2211 • **p19.402** Thanks. We prefer to keep it as it is. Both singular and
2212 plural forms appear to be grammatically correct [1]. According to
2213 this page, the singular form is used when the emphasis is on the
2214 variety itself, which matches our intention here.

2215 • **p19.409 these kinds of dynamics were or this kind of dy-**
2216 **namic was** Thank you, corrected. See line 479.

2217 [1] <https://grammarhow.com/there-is-a-variety-or-there-are-a-variety/>

2218 9. p20.447 lead to conserved (remove 'the') Thank you. We prefer to keep
2219 the word *the* here since we refer to specific dynamics and behavior of
2220 epileptor.

2221 10. p22.499 what is meant by 'mesoscopic'? We meant the microcircuit level of
2222 activity as opposed to cellular (microscopic) activity and larger network
2223 (macroscopic) activity. However, we agree that this distinction is not
2224 necessary and therefore we have removed "mesoscopic".

2225 11. p25.586 regulate, p25.572 and the associated

2226 • **p25.586 regulate** Thanks, corrected! Well spotted. See line 637.

2227 • **p25.572 and the associated** Thanks. Corrected. See line 643.

2228 12. p26.576 ...complexity is introduced by the fact...not only by neurological
2229 injury, but also from downstream Thanks. Corrected. See line 647.

2230 13. p28.632 and p29.653 too many commas Thanks. Corrected. See line 710
2231 and 732

2232 14. p29.655 evolution has removed Thanks. Corrected. See line 734.

- 2233 15. p29.667 using multiple models, p29.670 channel. Using multiple conductance-
2234 based... p29.671 remove comma
- 2235 • **p29.667 using multiple models.** Thanks for this suggestion. We
2236 prefer *populations of models* here, because population modelling is
2237 defined term.
 - 2238 • **p29.670 channel.** Thanks. Corrected.
 - 2239 • **Using multiple conductance-based..** We prefer to keep it for the
2240 same reason as above.
 - 2241 • **p29.671 remove comma** Unsure about this one. If we understand
2242 correctly, *Using ...* is an non-restrictive subordinate clause and thus
2243 should be separated by comma [1].
- 2244 [1] <https://www.grammarly.com/blog/subordinate-clause/>
- 2245 16. p33.748 for therapy... for basic research (remove 'the'), p33.751 evoke
2246 dysfunction in degenerate... Thanks. Both corrected as suggested.
- 2247 17. p34.783-787 remove "system's" or reword the sentence, awkward Thank
2248 you. We fully agree. It reads now *The larger the number of interacting*
2249 *mechanisms the more complex, flexible and robust the system becomes.* See
2250 line 911.
- 2251 18. p34.787 performing one function, A (redundancy), then... multiple struc-
2252 turally different... This sentence is still very unclear to me, consider re-
2253 moving this and the following sentence altogether– the point has been
2254 made already. Thank you. The two mentioned sentences are removed.
- 2255 19. I would also recommend removing many parenthetical extra words and
2256 just leaving them in the sentence, as this is a distracting way of writing

2257

for many of the cases in this paper. Thanks, we have tried to reduce the

2258

number of parenthetical extra words.

REVIEWERS' COMMENTS:

Reviewer #1 (Remarks to the Author):

I applaud the authors for their rigorous, thoughtful, and conscientious revisions. I feel the manuscript is significantly improved by the new material, and the authors justification for why they chose not to make other suggested changes is entirely reasonable. This review article will be a fantastic addition to the academic conversation regarding the complexities of epilepsy, and I am happy to endorse its publication.

Reviewer #2 (Remarks to the Author):

The authors have addressed the concerns adequately and the additions made on the basis of both reviewers' comments enhances the paper significantly and has clarified some of the points that were confusing in the original writing.

Overall, this is an interesting and timely review of some of the issue of degeneracy in epilepsy which may significantly affect the interpretation of results of many studies on the topic.